# Decomposing Complex Queries for Tip-of-the-tongue Retrieval

**Kevin Lin**♠ **Kyle Lo**♡ **Joseph E. Gonzalez** ♠ **Dan Klein**♠

♠University of California Berkeley ♡Allen Institute for AI

{k-lin,jegonzal,klein}@berkeley.edu kylel@allenai.org

## Abstract

When re-finding items, users who forget or are uncertain about identifying details often rely on creative strategies for expressing their information needs—*complex* queries that describe content elements (e.g., book characters or events), information beyond the document text (e.g., descriptions of book covers), or personal context (e.g., when they read a book). Standard retrieval models that rely on lexical or semantic overlap between query and document text are challenged in such retrieval settings, known as *tip of the tongue* (TOT) retrieval. We introduce a simple but effective framework for handling such complex queries by decomposing the query with an LLM into individual *clues*, routing those as subqueries to specialized retrievers, and ensembling the results. Our approach takes advantage of off-the-shelf retrievers (e.g., CLIP for retrieving images of book covers) or incorporate retriever-specific logic (e.g., date constraints). We show that our framework incorporating query decomposition into retrievers can improve gold book recall up to 6% absolute gain for Recall@5 on *WhatsThatBook*, a new collection of 14,441 real-world query-book pairs from an online community for resolving TOT inquiries. [1]

## 1 Introduction

Tip of the tongue (TOT) refers to the retrieval setting in which a user is unable to formulate a precise query that identifies a sought item, even if the user knows they've encountered this item before. For example, users searching for movies they watched or books they read long ago often resort to complex and creative queries that employ a diverse set of strategies to express information relevant to the sought item—high-level categories (e.g., topic, genre), content details from the movie or book (e.g., events, characters), references to personal context

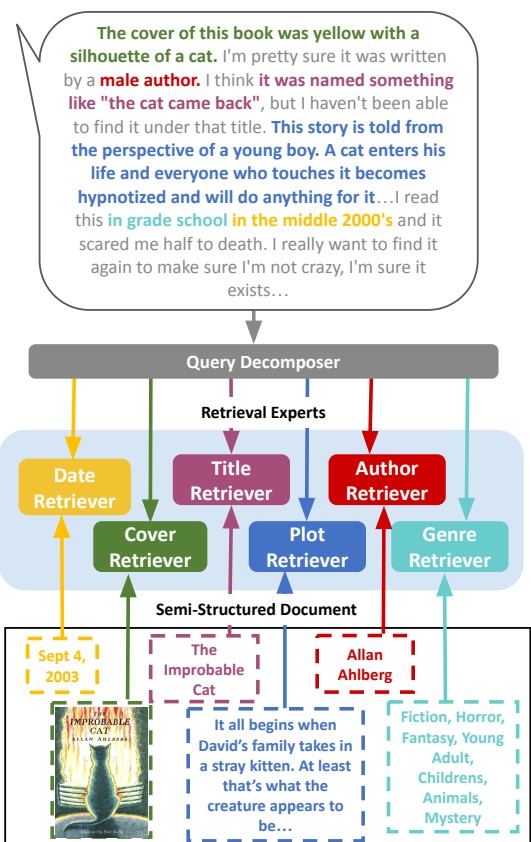

Figure 1: To resolve tip-of-the-tongue queries about books, our approach decomposes long, complex queries into subqueries routed to specific retrieval experts, each trained to handle a particular aspect of a book.

(e.g., when they last read the book), descriptions of extratextual elements (e.g., movie promotional posters, book covers), and more. In fact, in an annotation study of TOT queries for movies, Arguello et al. (2021) found over 30 types of informational facets that users may include when crafting queries. Figure 1 shows a TOT query and its corresponding gold book.

A key challenge in TOT retrieval is that queries are not just longer and more complex than those in popular retrieval datasets, but resolving them requires an enriched document collection since

---

[1]We release code and data at https://github.com/kl2806/whatsthatbook

query-document relevance can't necessarily be established from document content alone (see Table 1). For example, in Figure 1, the query's description of the book cover—*The cover of this book was yellow with a silhouette of a cat*—can be highly useful for identifying the book, but necessitates the book's representation to contain that information.

In this work, we present a simple yet effective technique for improving TOT retrieval: First, we decompose queries into individual subqueries or *clues* that each capture a single aspect of the target document. Then, we route these subqueries to expert retrievers that can be trained individually. Finally, we combine their results with those from a base retriever that receives the original query. Experiments show improvement in gold book recall over description-only retrieval baselines on a set of 14,441 real-world query-book pairs collected from an online forum for resolving TOT inquiries, complete with cover images and metadata.

## 2 Decomposing Queries

In this section, we describe our a simple but effective method for tackling long, complex TOMT queries. Given a collection of documents $d_1, \ldots, d_n$ and a textual query $q$, the TOT retrieval task aims to identify the sought document $d^*$. The input (raw) documents are semi-structured; each document $d$ contains metadata fields $d^{(1)}, \ldots, d^{(k)}$. In our the *WhatsThatBook* dataset, the documents are books that have the fields that correspond to plot, its publication year, an image of its book cover, title, genre, and author etc. Missing elements take on a default value (e.g., blank image, earliest publish date in overall book collection).

### 2.1 Method

First, the query decomposer takes a query $q$ and outputs a set of subqueries $q^{(1)}, \cdots, q^{(k)}$, for $k$ metadata fields. To do this, we use in-context learning with a large language model (LLM) to generate a subquery $q$ that is relevant to that field or optionally output the string "N/A" if the $q$ does not contain any relevant information to the field; this is repeated for each field and can be run in independently in parallel for each field.

We experiment with two prompting strategies, *extractive* and *predictive*. Extractive prompting aims to generate subqueries that are purely extractions from the original query. The LLM is instructed to output "N/A" if there is not information

relevant to the metadata field.

Predictive prompting aims to generate subqueries that are similar to the *content* of the metadata field. Note that the subqueries need not be extractions from the original query and can be inferred metadata or even hallucinations from the query decomposer. Using LLMs to generate subqueries affords us the ability to set the few-shot prompt generation targets to be *predictions*. This is important as the information in queries are rarely presented in a form amenable for matching with the corresponding document field. For example, books have publish dates, but queries will rarely mention these dates; instead, users may articulate personal context (e.g., "*I read this book in high-school around 2002-2005*"). Then to simplify the learning task for a date-focused retrieval expert, we might ask the LLM to predict a "latest possible publish date" (e.g., 2005).

See Table 2 for examples of extractive and predictive subqueries and Appendix A for examples of their prompts. In practice, we use GPT 3.5 (`gpt-3.5-turbo`) few-shot prompting with up to 8 in-context examples. Each field has its own prompt template and set of examples. Subqueries for different fields can be generated in parallel, as the they are independent of each other.

### 2.2 Retrieval Experts

We have retriever models, or experts, that specialize to specific field types. Let $R_1, \ldots, R_k$ represent these retrievers. Retrievers can be implemented as dense, sparse, or symbolic logic.

If a retriever requires training, we run the query decomposer over all query-document pairs $(q, d)$ in the training set. This produces effectively $k$ training datasets, where each dataset is comprised of a subquery and document-field pair. For example, field $j$ would have training dataset of examples $(q^{(j)}, d^{(j)})$.

At indexing time, each document's field is indexed according to the specifications of its retriever expert. For example, if the retriever is implemented as an embedding model, then that specific field is converted into an embedding. On the other hand, if the retriever is a sparse model, then a sparse index would be built using just that specific field's text.

At inference time, each retriever takes a subquery $q^{(j)}$ and retrieves documents from its associated index of fields.

| Dataset | Query Length | Lexical Overlap |
|---|---|---|
| MSMarco (Campos et al., 2016) | 7.68 | 0.55 |
| Natural Questions (Kwiatkowski et al., 2019) | 10.35 | 0.52 |
| BioASQ (Tsatsaronis et al., 2015) | 14.82 | 0.58 |
| TREC-COVID (Roberts et al., 2020) | 15.94 | 0.41 |
| SciFact (Wadden et al., 2022) | 19.52 | 0.50 |
| HotPotQA (Yang et al., 2018b) | 22.78 | 0.45 |
| TOMT (Bhargav et al., 2022) | 136.50 | 0.25 |
| *WhatsThatBook* | 156.20 | 0.19 |

Table 1: Tip of the tongue (TOT) queries are significantly longer while also having less lexical overlap with the gold document, compared with queries in popular retrieval datasets. Query length is number of BPE (Sennrich et al., 2016) pieces, averaged across examples. Lexical overlap is fraction of whole words in query that occur in gold passage(s), averaged across examples.

## 2.3 Implementation details

In practice, for titles, authors, plot, and genre, we use Contriever (Izacard et al., 2021), a state-of-the-art dense retriever.[2] For both models, we train for a total of 10,000 steps with a batch size of 16, learning rate of 1e-4. For titles, we finetune with 3,327 extracted subqueries. For our base retriever, we use the full training set of original book descriptions. For embedding search during inference, we use the `faiss` library and project all embeddings to 768 Johnson (2022).

For cover images, we use CLIP (Radford et al., 2021), a state-of-the-art image-text model that can be used for image retrieval by scoring matches between embedded images and their textual descriptions. Specifically, we finetune ViT-B/32[3] on 2,220 extracted subqueries using cross-entropy loss with batch size of 4, learning rate of 5e-5 and weight decay of 0.2 for 10 epochs with the Adam optimizer (Kingma and Ba, 2014). We select the model with the best top 1 retrieval accuracy on a validation set.

For publish dates useing the predictive prompting, we use a symbolic function that heuristically scores 0 if a book was published after the subquery date (i.e. predicted latest publish date) and 1 otherwise. If necessary, we heuristically resolve the subquery to a year.

## 2.4 Combining retrieved results

In this work, we restrict to a simple strategy of using a weighted sum of all $k$ retrieval scores across the $(q^{(j)}, d^{(j)})$. That is, the final score is:

$$s(q, d) = \sum_{j=1}^{n} w^{(j)} R_j(q^{(j)}, d^{(j)})$$

All documents are scored in this manner, which induces a document ranking for a given query $q$. We tune the weights $w_j$ on the validation set and select the weights that have the best Recall@5.

## 3 Datasets

**WhatsThatBook** We introduce the *WhatsThatBook* dataset consisting of query-book pairs collected from a public online forum on GoodReads for resolving TOT inquiries about books.[4] On this forum, users post inquiries describing their sought book and community members reply with links to books on GoodReads as proposed answers.[5] If the searcher accepts a book as the correct answer, the post is manually tagged as SOLVED and a link to the found book is pinned to the thread. For these solved threads, we take the original inquiry as our query $q$ and the linked book as gold $d^*$. At the end, *WhatsThatBook* contains 14,441 query-book pairs. Each query corresponds to a unique book. Finally, these books are associated with pages on GoodReads, which we used to obtain publication year metadata and images of book covers. We further collect 172,422 negative books (books that do not correspond to a gold query in our dataset) to make the setting more realistic, for a total of 186,863 books. To collect negative books, crawl books authored by crawling books by the authors of the positive books.

---

[2]https://huggingface.co/facebook/contriever

[3]https://huggingface.co/sentence-transformers/clip-ViT-B-32

[4]https://www.goodreads.com/group/show/185-what-s-the-name-of-that-book. We scraped data from February 2022.

[5]This is a simplification of community interactions. Threads also may include dialogue between original poster and members but this is beyond the scope of our work.

| Query-Document | Clue-Field | Clue-Field | Clue-Field |
| --- | --- | --- | --- |
| *Query:* I think I saw this in a used store once and I remember saying to my new husband "my daughter use to read that book to her little brother" and it's funny because on the outside cover is a little girl reading a book to her little brother. It's called.....my book, or my story, or something simple like that. It would be about 15 or more years old. The girl was blond and the boy brunet....I think!!!! Inside was the cutest little sentences and my kids use to do what each page said...thank you!! | *Extracted Title:* It's called.....my book, or my story, or something simple like that. | *Extracted Date Clue:* 15 years old (2006 or earlier) | *Extracted Cover:* The outside cover is a little girl reading a book to her little brother. |
| | | | *Actual Cover:* |
| *Description: Glossy pictorial hardcover no dust jacket. 2001 7.75x9.13x25. GUIDE FOR PARENTS WITH PICTURES, HOW TO TEACHING CHILDREN READING.* | *Actual Title:* My First Book | *Actual Date* First published September 1, 1984 | |
| *Query:* I read a book in 2008 or 2009 that was part of a series for young adults. It was fantasy, and about several families of witches and warlockes. The main character was named Holly. | *Extracted Genre:* for young adults, fantasy | *Predicted Plot:* Holly is a young witch who comes from a long line of magical families... | *Predicted Cover:* Young woman with long, curly hair holding a wand and surrounded by swirling colors and symbols |
| | | | *Actual Cover:* |
| *Description:* Holly Cathers's world shatters when her parents are killed in a rafting accident. She is wrenched from her home in San Francisco and sent to Seattle to live with her aunt, Marie-Claire, and her twin cousins, Amanda and Nicole... | *Actual Genre:* Fantasy, Young Adult, Paranormal, Romance, Fiction | *Actual Plot:* (see description) | |
| *Query:* Written in the 1990s, by [probably] one of those romance novelists (not Nora Roberts though). The cover was just a snowed-in cabin on a mountain side at night. In a nut shell - An escaped prisoner takes a woman captive and steals her car with her in it!...Any ideas what this might be called? I haven't forgotten it after all these years. | *Extracted Author:* by [probably] one of those romance novelists (not Nora Roberts though) | *Predicted Cover:* Dark, mysterious cabin obscured by falling snow with the silhouette... | *Extracted Cover:* The cover was just a snowed-in cabin on a mountain side at night. |
| | | | *Actual Cover:* |
| *Description:* A rootless foster child, Julie Mathison had blossomed under the love showered upon her by her adoptive family. Now a lovely and vivacious young woman... | *Actual Author:* Judith McNaught | *Actual Cover:* (see right) | |

Table 2: Query-document pairs, their generated sub-queries or *clues*, and corresponding gold document fields. Clues can be *extracted* directly from the query or *predicted* as a best-guess attempt to match the actual document field. Clues can be about the book's title, author, date, cover, genre, or general plot elements.

| Extract | Plot | Dates | Cover | Title | Genre | Author | Num Posts |
|---|---|---|---|---|---|---|---|
| 1 | 2657 (89.3%) | 138 (4.6%) | 43 (1.4%) | 80 (2.7%) | 39 (1.3%) | 18 (0.6%) | 2975 |
| 2 | 4630 (96.0%) | 2985 (61.9%) | 283 (5.9%) | 641 (13.3%) | 994 (20.6%) | 117 (2.4%) | 4825 |
| 3 | 3476 (97.6%) | 2759 (77.4%) | 856 (24.0%) | 1415 (39.7%) | 1841 (51.7%) | 342 (9.6%) | 3563 |
| 4 | 1821 (98.8%) | 1627 (88.3%) | 1092 (59.3%) | 1458 (79.1%) | 1064 (57.7%) | 310 (16.8%) | 1843 |
| 5 | 537 (99.4%) | 511 (94.6%) | 484 (89.6%) | 502 (93.0%) | 456 (84.4%) | 210 (38.9%) | 540 |
| 6 | 50 (100%) | 50 (100%) | 50 (100%) | 50 (100%) | 50 (100%) | 50 (100%) | 50 |
| **Total** | 13171 (95.5%) | 8070 (58.5%) | 2808 (20.4%) | 4146 (30.1%) | 4444 (32.2%) | 1047 (7.6%) | 13796 |

Table 3: Number of queries organized by results from running query decomposition with extractive prompting. Rows correspond to number of clues **Extracted** (between one and six). For instance, top row is queries with only a single extracted clue and bottom row is queries with all clue types found. **Num Posts** counts number of queries in each bucket. Cell counts correspond to number of times a given clue type was extracted in a post, and percentages normalize against **Num Posts**. For example, 61.9% of queries with two extracted clues have one "Date" subquery.

**Understanding *WhatsThatBook* queries.** To examine the information contained within queries, we analyze the results of query decomposition using extractive prompting (see Table 3). First, we find only 645 (4.5%) posts have no clue extracted (and thus aren't captured in the table total). Second, most posts have between one and three clues ($\mu$=2.33). Third, nearly every query contains some description of the book's plot elements. Beyond that, over half of the queries provide some description of temporal context surrounding the book. Queries containing descriptions of the author are rare, which is expected since knowing the author name would likely have allowed the user to find the book without seeking assistance. Given that, it's somewhat surprising that descriptions of titles occur almost a third of the time. Manually examining these, we find title clues are often uncertain guesses ("*I think the title might start with Nurse but I'm not sure*") or details that models are likely to struggle with ("*The title is made up of either two or four short words (maybe one syllable)*").

***Reddit-TOMT (Books)*** We additionally use the books subset[6] of the *Reddit-TOMT* (Bhargav et al., 2022) dataset, which includes 2,272 query-book pairs.[7] Queries can refer to the same book, leaving 1,877 unique books in this dataset.

**Experimental setup.** For the experiments in the rest of this paper, we split *WhatsThatBook* into

train ($n$=11,552), validation ($n$=1,444) and test ($n$=1,445) sets. By the nature of our dataset construction, the number of queries and books is equal. We use all 14,441 books, which are gold targets with respect to some query, as our full document collection for indexing. As for *Reddit-TOMT (Books)*, given its size we use the entire dataset as an additional test set.

## 4 Experiments

| Model | Top 5 | Top 10 | Top 20 | Top 100 |
|---|---|---|---|---|
| | | *WhatsThatBook* | | |
| BM25 | 8.5 | 12.5 | 16.2 | 22.5 |
| CLIP | 1.9 | 2.8 | 3.5 | 5.7 |
| DPR | 23.1 | 31.2 | 38.1 | 57.6 |
| ColBERT | 17.8 | 18.3 | 25.4 | 34.1 |
| Contriever | 26.5 | 33.5 | 40.3 | 61.3 |
| Contriever (E) | 26.7 | 33.4 | 41.8 | 60.5 |
| Contriever (P) | 29.3 | 35.5 | 42.1 | 61.7 |
| Contriever (H) | 25.0 | 34.1 | 40.2 | 60.4 |
| Contriever (Q) | 18.5 | 24.2 | 31.8 | 52.9 |
| **Ours (E)** | 26.6 | 34.1 | 40.2 | 60.4 |
| **Ours (P)** | **32.1** | **40.0** | **47.1** | **67.2** |

Table 4: Results on test set of *WhatsThatBook*. (Top) Baselines operate directly over queries and descriptions of the books. (Middle) We use the top performing model, Contriever, on the concatenated representations of subqueries (E-extractive, P-predictive, Q-rewritten queries) or the hypothetical document (H). (Bottom) Subqueries routed to individual retriever experts.

### 4.1 Baseline models

**Models that use original queries.** We evaluate our approach against several popular retrieval models that have been used as baselines for a range of other retrieval datasets (see Table 1). For text-only models—BM25 (Robertson and Walker, 1997; Robertson and Zaragoza, 2009), Dense Passage

---

[6]We note that the full *Reddit-TOMT* dataset also contains 13K queries matched with movies. To fully explore the capabilities of our approach, we restricted to the books subset specifically due to feasibility of obtaining images of book covers, while doing the same with movie posters was more difficult. We leave this to future investigations.

[7]We removed 47 query-book pairs for which the gold book did not have a GoodReads link, which was necessary for obtaining cover images.

| | | Reddit-TOMT | | |
|---|---|---|---|---|
| Model | Top 5 | Top 10 | Top 20 | Top 100 |
| Contriever | 42.8 | 51.1 | 58.6 | 77.7 |
| **Ours (P)** | **44.2** | **53.6** | **61.3** | **79.9** |

Table 5: Results on test set of Reddit-TOMT (Books). We report the best performing baseline (Contriever) using the original queries and the best performing model using our approach that uses the predicted subqueries (P).

Retrieval (DPR) (Karpukhin et al., 2020), Contriever (Izacard et al., 2021), and ColBERT (Khattab and Zaharia, 2020)—the document representation is simply the concatenation of all available document fields into a single text field. For our image-only baseline—CLIP (Radford et al., 2021)—the document representation is only the embedded book cover. All baselines receive the same input (full) query, and are finetuned with the same hyperparameters (§2.2) except using the full training set (as opposed to just examples with a successful subquery extraction).

**Models that use generated queries.** To evaluate the effect of generating subqueries independently without training specialized retrievers, we also train the top performing text-only model from the set of models with queries enriched with LLMs. We experiment with using the concatenation of all subqueries generated by the query decomposer as the query representation for both the extractive and predicted subqueries. To isolate the effect of generating separate subqueries, we also use LLM to generate a *single* rewritten query that is used as input to Contriever, by simply prompting LLMs to generate a single new query that better retrieves the desired item (prompt in Appendix A. In addition, we generate *hypothetical documents* and use these as the query, similar to Gao and Callan (2022) though we differ in that we train on these generations as queries while Gao and Callan (2022) restricted their use to the zero-shot setting.

## 4.2 Results

Table 4 shows the test results on *WhatsThatBook*. We use Recall@$K$ metric as our primary metric since each query has exactly one correct item.

**Contriever is the best-performing baseline model.** In this setting with low lexical overlap, we see that dense retrievers like DPR and Contriever outperform sparse retrievers like BM25.

Without extracting clues about the book cover, using CLIP on its own is not effective, likely due to its limited context window.[8]

**Query decomposition improves performance.** When using the rewritten queries without training individual expert retrievers, we observe that concatenating the predicted clues improve the model. However, both generating rewritten queries and entire hypothetical documents perform worse than just using the original document as input. Our approach to decompose queries and route clues to specialized retrievers improves performance (ranging from +6 to +7 Recall@$K$ across all cutoffs) over the next best baseline retriever. Table 6 shows the results for each individual expert retriever on *WhatsThatBook* and the books subset of TOMT.

**Performance degrades predictably with corpus size** To see how well the performance scales as more documents are added, we also report the performance of the base performing baseline and our model as we add more documents to the corpus. We evaluate the performance for different corpus sizes by varying the number of negative books added to the corpus. Figure 2 shows that for both our decomposition and the baseline, the performance drops significantly as more negative books are added. There is a sharp decline of performance when negative books are first introduced (between corpus size 1 and 2). Since the negative books are collected from the authors of the positive books, they may be more difficult to distinguish than a random book. As more documents are added, there is a marginal decrease in performance smoothly. Furthermore, our decomposition method performance better than the baseline for different corpus sizes and has slightly less performance degradation as more documents are added to the corpus.

**Metadata fields exhibit long-tailed behavior.** The query decomposer generates a plot subquery in at least 90% of the queries for both *WhatsThatBook* and TOMT. Dates occur in a large proportion of the queries, but are not specific enough to be effective identifying the book. While subqueries for the author appear very infrequently, when they do appear, they are much more effective than more generic subqueries such as dates or genres. Images

---

[8]We pass the full query into CLIP and allow for truncation to happen naturally. This is a big issue with CLIP, which supports a narrow query length; hence, motivating our approach to extract *clues* about book covers from the full query.

| Subquery | WhatsThatBook | | | | | Reddit-TOMT (Books) | | | | |
| | Top 5 | Top 10 | Top 20 | Top 100 | % not N/A | Top 5 | Top 10 | Top 20 | Top 100 | % not N/A |
|---|---|---|---|---|---|---|---|---|---|---|
| plot (E) | 29.0 | 36.4 | 44.9 | 64.2 | 90.4 | 45.5 | 56.2 | 61.0 | 76.0 | 94.3 |
| plot (P) | 27.9 | 34.7 | 42.4 | 61.7 | 100 | 43.2 | 50.5 | 58.8 | 77.1 | 100 |
| dates (E) | 1.3 | 2.1 | 3.4 | 6.0 | 54.8 | 0.0 | 1.0 | 2.2 | 2.8 | 59.8 |
| dates (P) | 0 | 0 | 1.1 | 2.3 | 100 | 0.0 | 1.8 | 2.3 | 4.4 | 100 |
| cover (E) | 5.3 | 6.7 | 7.7 | 14.1 | 21 | 8.6 | 11.6 | 15.5 | 17.7 | 35.8 |
| cover (P) | 3.1 | 4.4 | 5.6 | 7.2 | 100 | 5.8 | 8.4 | 10.0 | 13.4 | 100 |
| title (E) | 11.7 | 12.9 | 14.6 | 19.8 | 29.0 | 13.4 | 17.5 | 22.2 | 37.4 | 43.9 |
| title (P) | 4.3 | 5.7 | 8.2 | 13.4 | 100 | 14.3 | 17.2 | 21.0 | 35.5 | 100 |
| genre (E) | 2.7 | 3.4 | 5.9 | 17.4 | 28.2 | 1.3 | 4.4 | 5.2 | 28.8 | 26.8 |
| genre (P) | 3.8 | 6.8 | 10.9 | 24.4 | 100 | 6.7 | 8.6 | 12.3 | 25.5 | 100 |
| author (E) | 6.0 | 8.0 | 8.0 | 9.1 | 6.9 | 7.1 | 8.2 | 9.6 | 9.6 | 9.9 |
| author (P) | 0 | 1 | 1.1 | 3.2 | 100 | 1.9 | 3.0 | 4.7 | 14.5 | 100 |

Table 6: Results of individual retrieval experts on the test set of *WhatsThatBook* and *Reddit-TOMT (Books)*. E and P indicate extractive and predictive prompts. Extractive prompts scores are calculated only over the "N/A" queries.

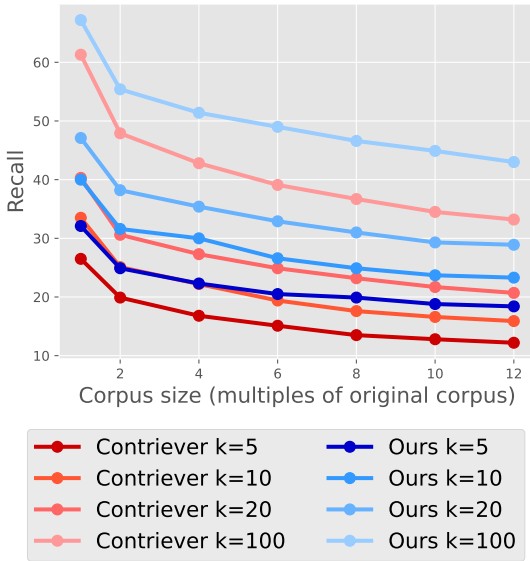

Figure 2: Recall at k of the top performing baseline (Contriever) and our decomposition baseline for vary amounts of negatives documents added to the corpus.

are much more effective when using decomposition compared to the image only CLIP baseline, as the image retriever model is able to access the visual part of the text

**Predictive prompting performs better than extractive prompting.** Overall, we find that predicting subqueries is more effective than extracting. Subqueries are generated for every query, thus, there is more data for the specialized retriever experts to train on, compared to extractive clues where some retrievers only have a small fraction of the dataset for finetuning. Moreover, the predictive clues allows the LLM to make inferences based

on information that is not explicitly present in the query. For example, the LLM is able to make inferences about the genre effectively based on the full query even when the user does not explicitly discuss the genre. Another benefit of the extractive clue is that the subqueries are more grounded in the original query.

**Trade-off exists between generating subquery and individual retriever expert performance.** Between the prompting strategies, we find that there is often a trade-off between how selective the query decomposer is with generating a subquery for a metadata field, and the effectiveness when generated. For most of the metadata retrieval queries, the extractive prompting approach is slightly more effective than predictive prompting on the examples that it does not predict "N/A" on.

## 5 Error Analysis

We sample 50 predictions where the top performing model where the model fails to get the book correct in the top 5 and categorize the types of errors made by the model in Table 7. The most common kind of error is failure to model the relationship between the document and query, which happens in instances where there may be events that dense models fail to capture indicating that there is still headroom on the task. Moreover, documents are sometimes brief or written in a style that is not similar to other descriptions. Lastly, because users are recalling memories, some of these can be *false memories* that lead to errors in retrieval.

| % | Error type | Query | Document |
|---|---|---|---|
| 54% | Fail to Query-Doc Relationship t | "I think this was a teen book. Don't remember Author or character names. All " I remember is that the girl loses her memory, it is not " Memoirs of a teenage amnesiac". It was fiction book and it wasnc̃omic or manga.I think the girl was involved in a car accident where someone hit her while she was walking? Apparently she was quite wild and broke instruments and was quite hated before she had the accident or something like that for things she done "to people, but she cant remember any of it. ... | *Title* Kat Got Your Tongue' *Plot* After a terrible car accident, Kat wakes up with no idea who she is, and no memory of anything before the crash. She "doesnt even recognize her mum, much less her friends from " school—Poppy, Jade, and the mysterious Tina. Only after she finds her old diary—written in a voice no longer her own—does Kat begin to discover the terrible secrets of her previous life.This incredib After a terrible car accident, ... |
| 28% | Document Representation Insufficient | I read this book many years ago and it was an older book so I do not know the publication year. What I remember is a mother with two daughters that were young ladies possibly and their late teens early twenties. The mother "favored the one daughter named Charlotte... | *Title* The Daughters of Ardmore Hall *Plot:* An unbalanced woman seeks to destroy her daughters. |
| 18% | Query Contains Error | Fiction, read in the 70s by my mother who thinks it was probably written in that time period too. Book is about a dying woman remembering her childhood during a war in (probably South) Africa.Title might have something to do with a dragon or a mosquito coil :) | *Title:* Moon Tiger *Plot:* The elderly Claudia Hampton, a best-selling author of popular history; lies alone in a London hospital bed. Memories of her life still glow in her fading consciousness, but she imagines writing a history . ... |

Table 7: Error types from the top performing model. Book representation are simplified here without all metadata for space constraints.

## 6 Related Work

**Dense methods for document retrieval.** Document retrieval has a long history of study in fields like machine learning, information retrieval, natural language processing, library and information sciences, and others. Recent years has seen the rise in adoption of dense, neural network-based methods, such as DPR (Karpukhin et al., 2020) and Contriever (Izacard et al., 2022), which have been shown can outperform sparse methods like BM25 (Robertson and Walker, 1997; Robertson and Zaragoza, 2009) in retrieval settings in which query-document relevance cannot solely be determined by lexical overlap. Researchers have studied these models using large datasets of query-document pairs in web, Wikipedia, and scientific literature-scale retrieval settings (Campos et al., 2016; Sciavolino et al., 2021; Roberts et al., 2020). Many retrieval datasets have adopted particular task formats such as question answering (Kwiatkowski et al., 2019; Yang et al., 2018b; Tsatsaronis et al., 2015) or claim verification (Thorne et al., 2018;

Wadden et al., 2022). We direct readers to Zhao et al. (2022) for a comprehensive, up-to-date survey of methods, tasks, and datasets.

**Known-item and TOT retrieval.** Tip of the tongue (TOT) is a form of known-item retrieval (Buckland, 1979; Lee et al., 2006), a long-studied area in the library and information sciences. Yet, lack of large-scale public datasets has made development of retrieval methods for this task difficult. Prior work on known-item retrieval focused on constructing synthetic datasets (Azzopardi et al., 2007; Kim and Croft, 2009; Elsweiler et al., 2011). For example, Hagen et al. (2015) released a dataset of 2,755 query-item pairs from *Yahoo!* answers and injected query inaccuracies via hired annotators to simulate the phenomenon of *false memories* Hauff and Houben (2011); Hauff et al. (2012), a common property of TOT settings.

The emergence of large, online communities for resolving TOT queries has enabled the curation of realistic datasets. Arguello et al. (2021) categorized the types of information referred to in TOT queries

from the website *I Remember This Movie*.[9] Most recently, Bhargav et al. (2022) collected queries from the *Tip Of The Tongue* community on Reddit[10] and evaluated BM25 and DPR baselines. Our work expands on their work in a key way: We introduce a new method for retrieval inspired by long, complex TOT queries. In order to test our method on a large dataset of TOT queries, we collected a new dataset of resolved TOT queries such that we also had access to metadata and book cover images, which were not part of Bhargav et al. (2022)'s dataset.

**Query Understanding and Decomposition.** Our work on understanding complex information-seeking queries by decomposition is related to a line of work breaking down language tasks into modular subtasks (Andreas et al., 2016). More recently, LLMs have been used for decomposing complex tasks such as multi-hop questions into a sequence of simpler subtasks (Khot et al., 2022) or smaller language steps handled by simpler models (Jhamtani et al., 2023).

Related to decomposition of long, complex queries for retrieval is literature on document similarity (Mysore et al., 2022) or query-by-document (QBD) (Yang et al., 2018a). In these works, a common approach is decomposing documents into sub-passages (e.g. sentences) and performing retrieval on those textual units. The key differentiator between these works and ours is that document similarity or QBD are inherently symmetric retrieval operations, whereas our setting requires designing approaches to handle asymmetry in available information (and thus choice of modeling approach or representation) between queries and documents. In this vein, one can also draw parallels to Lewis et al. (2021), which demonstrates that retrieving over model-generated question-answering pairs instead of their originating documents can improve retrieval, likely due to improved query-document form alignment. In a way, this is similar to our use of LLMs to generate clues that better align with extratextual document fields, though our work is focused on query-side decomposition rather than document-side enrichment. More recently, Wang et al. (2023) propose using LLMs for decomposing different facets of complex queries for scientific documents.

---

[9]https://iremberthismovie.com/
[10]https://www.reddit.com/r/tipofmytongue/

# 7 Conclusion

We study tip of the tongue retrieval, a real-world information-seeking setting in which users issue long, complex queries for re-finding items despite being unable to articulate identifying details about those items. We introduce *WhatsThatBook*, a large challenging dataset of real-world TOT queries for books. We also introduce a simple but effective approach to handling these complex queries that decomposes them into subqueries that are routed to expert retrievers for specialized scoring. Our simple framework allows for modular composition of different retrievers and leveraging of pretrained models for specific modalities such as CLIP for document images. We experiment with different subquery generation strategies and find that generating predictions of document fields is more effective. We observe improvements up to 6% absolute gain over state-of-the-art dense retrievers for Recall@5 when incorporating query decomposition into existing retrievers.

# Limitations

Given our proposed method is designed to handle TOT queries, there is an implicit assumption that the document collection covers the sought item with high probability. That is, a system tailored for TOT retrieval is unlikely to perform well if its index is missing too many books. While our dataset is large-scale, one limitation is that even a corpus of 187k documents does not cover the sought document sufficiently. Consider, for instance, the total number of books available on GoodReads (over 3.5B as of time of writing). Another limitation of our method is that the overhead for tackling another domain with this technique is non-trivial as the prompts and few-shot examples may not be directly transferable. We believe a promising avenue for future work is reducing the effort needed to bootstrap the design and training of retrieval experts and incorporating them into a query decomposer.

# 8 Acknowledgements

We thank the members of the Berkeley NLP group and the anonymous reviewers for their insightful feedback. This work was supported by a grant from DARPA SemaFor, and gifts from sponsors of Sky Computing and BAIR. The content does not necessarily reflect the position of the sponsors, and no official endorsement should be inferred.

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

# A  Prompts

## Extractive Prompt

---

You are a utility built to take in forum queries from users looking for books and output the aspect that is about the cover. If there is not enough information, then output N/A. Do not guess and only output text that was in the
 query. Here are some examples:
Question: Hi there, I read this book in highschool around 2002-2005. From what I remember, the main character is nicknamed "Mouse" and she rides a big chestnut horse in jumper shows. I think this book may have been Australian. The cover just showed a chestnut horse and rider in mid jump. I think the title was one word--it may have been the name of the horse.I cannot remember the name or the author of this book. I have googled everything I can think of but I cannot to find this book and its driving me crazy! I'd be grateful for any help on this! Thank you! Cover Clue: The cover just showed a chestnut horse and rider in mid jump.
Question: So I read this book somewhere between 2008-2010 when I was in elementary school or middle school. It was about a girl who lived by the sea, and started out with her having dinner at this boarding house she lived in. And something happened (she received a fruit or something else that was banned in her town) and she had to hide it. Then, the landlady of her house got mad when she found out. The MC went down to the beach and saw this giant walrus thing (I can't remember clearly but I think there was also a guy who rode the walrus but maybe not) and the MC got on the walrus and they rode away to this magical land. I don't remember much else, except the story had something to do with fruits for some reason, and the characters mother was likely from this magical land. Also, in the end, the main character brings Color and happiness back to her seaside village and I think there's something else to do with strawberries. Please help! I've been searching for this book for years! Thanks so much <3 Cover Clue: N/A
Question: I remember reading this around 2008/2009. It was about a girl being prettier than her mother, and when the mother gets jealous she sends her daughter to a type of boarding school for people who are well-known (Like nobles and princesses. She may be royalty). The cover was purple and captivating,( which was why I picked the book to read ^^) with a girl's face on it. However, I'm not sure if the cover really had a face on it, or if it was completely purple.(view spoiler)[I believe that at the end, a minion of the girl's mother helped the girl get rid of her mother. Maybe it was because the mother tried to kill her daughter for being beautiful, especially since she was getting older. (hide spoiler)] Please help me find it. It was the first book that got me to love reading at that time. :) Cover Clue: The cover was purple and captivating,(which was why I picked the book to read ^^) with a girl's face on it. However, I'm not sure if the cover really had a face on it, or if it was completely purple.
Question: I probably read this book in the late 90s, early 2000s, and cannot find the title anywhere. The main character is an outcast in her high school who lives in a trailer park near a cemetery. From what I remember she was walking through the cemetery trying to clear her mind when she wondered upon a funeral that had very few people at the funeral. After that she starts attending funerals on the weekend. She even has an outfit she wears just to go to these funerals. Cover Clue: N/A
Question: hi im looking for a book there was two girls. the older one liked dumpster diving and had a duck shaped jar of memories that she could go into. the younger one was blonde and liked buying shirts from a thrift store and writing on them, and she c o u l d n t remember a n y o n e s faces. the younger one tried to mail her mom a coconut and the mailman was the only face she could remember. there was also a woman in a hospital who only said  red   s h o e s  and the cover was green or blue. was read a couple years ago so probably 2017-2019ish? Cover Clue: the cover was green or blue.
Question: This book was blue (if that helps). And I think it had a mailbox on the front of it. There were three girls and they hated their rival school. So to get the two schools to be friends they made everyone send letters to another person in the school. One girl has a guy who wanted to impress a girl at his school so him and the girl he was sending letters went on on "Practice Dates" And they ended up liking each other :) The other girl likes playing "Games" with her letter guy... i dont really remember what happened with them. And the third girl had a jerk as her letter guy.. I dont really remeber what happened with him either.If anyone knows the name of the book I'm looking for would you please let me know? :) Thanks so much :) :) :) Cover Clue: This book was blue (if that helps). And I think it had a mailbox on the front of it.
Question: So I do not remember but the title, author, or cover, but I remember a bit of the plot. I believe it starts out with the main characters set up on a date. They  d o n t  know each o t h e r s  names, but their date consists of a sexy   get   to know  y o u /photo shoot where  t h e y r e  not allowed to have sex but they do anyway. She gets pregnant but  c a n t  find him after. Flash forward a bit, she works for her  f a t h e r s  large New York office and they make a deal to with with her baby  d a d d y s  office. He finds out  s h e s  pregnant and so the story goes. I hope this is enough to go on and someone recognizes it. Thank you! Cover Clue: N/A

Now here is the example, remember not to add in additional information that's not in the question.Question: I read this book back around 2008 (I think) can't remember author but read two of her works 1) is about this girl who falls in love with a guy who is actually a dragon, he takes her back to his homeland and I think he is injured, there are other dragons there and his relatives are also dragons author gives very vivid imagery I believe there were types of dragons like fire and ice ones... 2) the second book ( warning:spoilers ahead) is about this girl who has a sister named rose (not too sure about name) who is already engaged to some local village boy, but she falls in love with this cold man who is like "ivy" and they escape together for a while but she returns and the girl also loves him but he eludes her grasp, I remember the end she talks about the imagery on the wall of roses and ivy intertwining together... Any help much appreciated!!! :) Cover Clue:

## Predictive Prompts

---

You are a utility for guessing titles of books. Given the book described below, what is a possible title for the book? Only return one predicted title without any extra text. Even if you're unsure, try to come up with something.\n\nDescription: {}

---

You are a utility for guessing author names. Guess a possible author for the book described below. Don't worry if you're unsure. Only return the name and a short explanation of why.\n\nDescription: {}

---

You are a utility for categorizing books. Given the book description below, generate several possible genre tags for the book. Try to have a diversity of coarse and fine-grained genres. Don't generate more than five genres.\n\nDescription: {}

---

You are a useful tool for generating ideas for cover art. Write 1 or 2 sentences depicting what a the cover of a book might look like based on the description below. Stick to only one idea.\n\nDescription: {}

---

You are a writing tool for generating ideas. Write a possible plot synopsis for a book based on the description below. Don't include the title or author.\n\nDescription: {}

## Query Rewrite Prompt

---

"You are a utility for helping users find books. Given the user's book description below, generate a query that a user can copy-paste into a book database to find the book. The query should focus on the important aspects of the book that will help the database locate it. These can be keywords about the book's title, author, genre, year, or distinguishing character or plot elements.\n\nUser's Query: {}\nDatabase Query: "