# OpenReview forum: "Decomposing Complex Queries for Tip-of-the-tongue Retrieval"
_EMNLP/2023/Conference — EMNLP 2023 Findings_

### Official Review · Reviewer_fUsA · 2023-07-26

**Soundness:** 3

**Excitement:**

3: Ambivalent: It has merits (e.g., it reports state-of-the-art results, the idea is nice), but there are key weaknesses (e.g., it describes incremental work), and it can significantly benefit from another round of revision. However, I won't object to accepting it if my co-reviewers champion it.

**Paper Topic And Main Contributions:**

This paper tries to solve an interesting NLP task, known as tip of the tongue (TOT) retrieval, by utilizing LLM for query decomposition. Then it uses different specialized retrievers for decomposed queries and ensembles the results. TOT is a NLP application problem that's interesting and worth studying, and this paper proposes a easy and tricky method applying LLM for solving this problem. The main contribution of this paper contains two parts. It provides a new TOT dataset and proposes a simple yet effective method for solving it.

**Questions For The Authors:**

Question A: Why uses multiple recall? Is there any experiments showing that in TOT multiple recall is better than a unified multi-modal and multi-field model?

Question B: Why "generating entire hypothetical documents performs worse than just using the original document" can prove "query decomposition improves performance"? I think only answer to Question A can prove this conclusion. What's more, generating hypothetical documents is obviously not a good idea since you can not finetune GPT3.5 for such generation.

Question C: Comparing the results of Contriever, Contriever (E), Ours (E) and Ours (P) in Table 4, it seems that decompose and extracted clues does trivial improvements. The main contribution comes from the predictive clues. Does that means the performance mainly comes from the knowledge of LLM?

Question D: If we set all the combination weights to be the same, what will the results be?


**Reasons To Accept:**

Provides a new TOT dataset.

Based on its related work (if comprehensive), it seems that this work is the first NLP solution that specializing for TOT problem.

The solution is simple and quite straightforward.

Detailed analysis on the data.


**Reasons To Reject:**

Lack of experiments
    The author introduces two datasets, WhatsThatBook and Reddit-TOMT(Books), but only gives results on WhatsThatBook. In the paper only the individual retrieval results of Reddit-TOMT(Books) are shown.
    Lack of ablation study. At least two ablation experiments should be conducted to verify the effectiveness of 1) LLM and 2) decompose, such as 1) replacing LLM with other query decompose methods or 2) use LLM to rewrite queries without decomposing. The author gives results of single expert on LLM decomposed queries (but concatenated), which is unclear for ablation.
    Lack of baselines. It uses BM25, DPR and ColBERT as baselines besides single expert.
        No model after 2021 are introduced as baselines.
        What's more, they only compared with single-modal baselines. I think it would be more convincing to add a unified multi-modal baseline model to prove the effectiveness of decomposing.
        The author treat all fields as text and concatenate them together for baselines, which seems to be unfair. It is more reasonable that the passage encoder in baseline models treat different fields as different kinds of features, such as making date and genre as id-like features.
        Lack of reports on the accuracy of LLM extracting/generating clues


Due to the lack of experiments and analysis described above, I'm not sure which design makes the real contribution to the effectiveness of this method, decompose or LLM?

If it's decompose, I'm not sure why bother using multiple experts and perform multiple recall and combine them together? Decomposing and Combination is not a method that generalizes well in many scenarios. In the paper the author uses a validation datasets to pick the best combination weights which is tricky and may lead to overfitting problem (if the distribution of validation datasets is shifted compared with test set, the results will be poor).

If it's LLM, then maybe we can just use LLM to rewrite the queries (a prompts such as " You are a utility built to taking forum queries from users looking for books and output six kinds of clues, including dates, cover description, title description, plot summary, authors and genres."). Then this problem degenerates to a prompt engineering for query rewrite.

**Reproducibility:**

4: Could mostly reproduce the results, but there may be some variation because of sample variance or minor variations in their interpretation of the protocol or method.

**Reviewer Confidence:**

4: Quite sure. I tried to check the important points carefully. It's unlikely, though conceivable, that I missed something that should affect my ratings.

**Typos Grammar Style And Presentation Improvements:**

line 172: useing -> using

line 73: we describe our a simple -> we describe our simple

---

> ### Author Rebuttal · Authors · 2023-08-29
>
> We thank the reviewer for the detailed feedback. We are glad that the reviewer found our approach simple and novel, and our data analysis to be valuable. We address each of the reviewer’s concerns below.
>
> Concern: Omission of TOMT experimental results.
> Response: We omitted the TOMT due to space constraints and we will add them in the final version of the paper in the additional space. In particular, the decompose-and-retrieve with predicted clues has the following results:
>
> Recall at 5 = 44.2
>
> Recall at 10 = 53.6
>
> Recall at 20 = 61.3
>
> Recall at 100 = 79.9
>
> Compared to the Contriever baseline:
>
> Recall at 5 = 42.8
>
> Recall at 10 = 51.1
>
> Recall at 20 = 58.6
>
> Recall at 100 = 77.7
>
> Concern: Choice of baseline retriever models.
> Response: We clarify that we are proposing a framework, decompose-and-retrieve using LLMs for tackling complex queries, rather than a new retriever model. Thus, our primary goal is to show that widely-used and competitive retriever models function well as a component within this framework. As such, we believe we’ve chosen a reasonable set of models — BM25, CLIP, DPR, ColBERT, Contriever — and having found that Contriever is the best of them, used it as the basis within our decompose-then-retrieve framework. Recent work that incorporate retrievers also often use Contriever as the representative dense retriever [1] or show that Contriever still is the state-of-the-art in supervised adaptation settings [2].
>
> Concern: Lack of unified/multimodal baseline models.
> Response: We agree this is an interesting idea and believe it would be an exciting direction for future work. Our contribution comes from studying complex queries, identifying the intuition that they reference multiple document (multimodal) attributes, and operationalizing a decomposition framework that capitalizes on this observation. Of course, while our approach is not the only approach, we believe it is a reasonable one as a starting point of research for this relatively understudied retrieval setting.
>
> Multimodal models would be another such approach that capitalizes on our observations, but we believe it would not be as a baseline. For example, one would need to figure out how to handle cases in which the query does not reference the book’s cover or when the book’s cover is missing. Working through such considerations would likely be considered its own contribution rather than a baseline model, which we hope to see in future works. Finally, to reiterate, our contribution is that of a framework that leverages multiple retrievers, and not a new retriever model itself; we hope that future development of multimodal models could be used in conjunction with our framework, similar to how we are able to leverage CLIP.
>
>
> Question A: The reviewer asks why not use a single unified retriever instead of multiple retrievers?
> Response: See above discussion around concerns of our baseline retrievers. We hope our clarification around our contribution as a framework and not an individual model helps address this.
> Question B: The reviewer questions the use of baselines like Contriever (H), which generate a hypothetical document for use in retrieval, as evidence in favor of our decomposition framework.
> Response: Our framework is two-fold: (1) includes using LLMs to create the sub-queries, and (2)  individual training of specialized retrievers.
>
> Our baseline approaches Contriever (E) and Contriever (P), which concatenate generated sub-queries into one query for one retrieval operation, help us evaluate (1) without (2). Contriever (H), or generating entire hypothetical documents, is one strong baseline that helps us evaluate them, as recent work has shown that generating hypothetical documents can be very effective at improving on state-of-the-art retrievers, even zero-shot [3]. If Contriever (H) had been better, then it would have indicated our decomposed information wasn’t as useful as another powerful yet simple generative technique. But experiments in Table 4 verified that our decomposed information was indeed better.
> Based on the reviewer’s concerns though, we have added one more generative baseline — Use an LLM to generate a rewritten query that’s used as input to Contriever, rather than a hypothetical document. We call this Contriever (Q). If this works, then that would be another sign that explicit decomposition is not useful compared to another simple LLM approach. The results are:
>
> Recall at 5 = 18.5
>
> Recall at 10 = 24.2
>
> Recall at 20 = 31.8
>
> Recall at 100 = 52.9
>
> which are quite a bit lower than Contriever (H) and others. We hope this helps convince the reviewer that decomposition is indeed an important aspect of tackling these complex, TOT queries.
> Question C: The reviewer is wondering whether the close results of Contriever, Contriever (E), Ours (E), and much better results of Ours (P) indicates that performance is being driven by LLM knowledge rather than decomposition?
> Response: Indeed, the difference between Extraction vs Prediction is one of our main findings. That is, purely representing the extracted clues as-is is insufficient, and that an LLM’s predictions (which make use of pretrained knowledge) are important for this task.
> That being said, we still believe there is evidence that decomposition is valuable. First, Contriever (P), in which the original query is decomposed into predicted sub-queries which are concatenated as a single input into Contriever, outperforms models that don’t involve decomposition (Contriever). Second, as in our response to Question B above, baselines that involve providing explicitly decomposed queries — Contriever (P) — outperform baselines that don’t — Contriever (H) and the newly added Contriever (Q).
>
> Question D: The reviewer asks what the results would look like if the combination weights are set to the same.
> Response: If we set the combination weights to be the same, we get:
>
> Recall at 5 = 19.7,
>
> Recall at 10 = 26.6
>
> Recall at 20 = 33.9
>
> Recall at 100 = 55.2
>
> Which is much lower than the scores of the tuning the combination weights. This shows that tuning the combination weights is crucial, as the retrievers are trained independently and calibrated to each other.
>
> Overall, we thank the reviewer for their thoughtful comments. We will make use of the additional page to add these additional results and clarify more the nature of our contributed framework relative to other retrieval modeling advances. We hope this increases the reviewer’s confidence in our work.
>
> [1] https://arxiv.org/pdf/2305.14795.pdf
>
> [2] https://aclanthology.org/2022.emnlp-main.149.pdf
>
> [3] https://arxiv.org/abs/2212.10496

---

### Official Review · Reviewer_Tnm2 · 2023-08-03

**Soundness:** 3
**Typos Grammar Style And Presentation Improvements:** Line 290

**Excitement:**

3: Ambivalent: It has merits (e.g., it reports state-of-the-art results, the idea is nice), but there are key weaknesses (e.g., it describes incremental work), and it can significantly benefit from another round of revision. However, I won't object to accepting it if my co-reviewers champion it.

**Paper Topic And Main Contributions:**

Authors present a methodology for tip of the tongue (TOT) retrieval by decomposing long and complex queries. To this aim, they introduce WhatsThatBook, a large dataset of real-world TOT queries for books.

Experiments based on extractive and predictive strategies are run. According to the presented results, incorporating query decomposition into existing retrievers shows an improvement of the state-of-the-art results gained by dense retrievers.



**Reasons To Accept:**

The topic of TOT retrieval is interesting and quite challenging.

The release of a real-world TOT dataset represents a plus.



**Reasons To Reject:**

Despite the results, I am not completely convinced by the methodology.

No particular strategy is implemented for decomposition, which seems to happen just through the top-down identification of some metadata categories in order to extract the corresponding content from the queries.

No additional (e.g., linguistic or context) information or differences among the metadata categories are factored into the decomposition process or clue retrieval, which makes me question the scalability of the method.

Another concern regards the prompting, as I am not sure that the applied prompts are sufficient for the purpose. The lack of comparison with the results of non-decomposed queries and with other prompts including  linguistic, language-related, and/or categories aspects limits the result evaluation and the paper contribution to explaining LLMs.



**Reproducibility:**

4: Could mostly reproduce the results, but there may be some variation because of sample variance or minor variations in their interpretation of the protocol or method.

**Reviewer Confidence:**

4: Quite sure. I tried to check the important points carefully. It's unlikely, though conceivable, that I missed something that should affect my ratings.

---

> ### Author Rebuttal · Authors · 2023-08-29
>
> We are glad that the reviewer found the topic of TOT retrieval interesting and our dataset to be a valuable contribution for further study on this topic.
>
> The reviewer expressed concern around our decomposition method. To reiterate, our contribution is our approach of using an LLM to decompose complex and lengthy TOT queries into subqueries that current retrievers can better handle.
>
> Our contribution is very much motivated by our observation that current models perform poorly on TOT queries, and our desire to better understand what makes TOT queries challenging. In collecting and studying this dataset, we find that they’re composed of various clues that each reference some specific attribute of the sought document. Based on this observation, we form the intuition that decomposition into more modular retrieval settings (i.e. sub-queries, specialized retrievers) may be a useful retrieval strategy. And we’ve empirically demonstrated (Table 4) an implementation of this intuition using LLMs that results in performance gains. Indeed, this specific implementation relies on availability of metadata, but we believe leveraging this structure is a reasonable first step when tackling a nascent, challenging problem.
>
> We fully agree with the reviewer that top-down identification of metadata categories is only one way of implementing such a decomposition, and that exploring more sophisticated methods that can automatically induce decompositions when such metadata structure is not available is an interesting open research question.
>
> We hope that our work demonstrating the potential for decomposition is a reasonable starting place for research in this exciting area. We hope our dataset will promote future work, especially that which tackles better ways of inducing the decomposition when metadata structure is not available.

---

### Official Review · Reviewer_hexP · 2023-08-05

**Soundness:** 4

**Excitement:**

4: Strong: This paper deepens the understanding of some phenomenon or lowers the barriers to an existing research direction.

**Paper Topic And Main Contributions:**

In this paper the authors propose a dataset for information retrieval, WhatsThatBook. The dataset focuses on searching the book using user's description as the query. Different from other IR datasets, the queries of WhatsThatBook are long (156 tokens per query on average), and the queries have low lexical overlap with the target documents. The authors build this dataset by scraping an online forum. Eventually the authors collected about 14k query-book pairs.

To solve this problem, the authors proposed a retrieval-by-decomposition approach. Specifically, each long query is firstly decomposed to sub-queries w.r.t. different fields (e.g., date, author, plot, etc.,), then each sub-query is issued to individual retrievers. The sub-query generation is done with GPT-3.5 and few-shot prompts. Finally the scores of different retrievers are aggregated. The authors compare the proposed retrieval approach with a few classical IR baselines, such as BM25, DPR and contriever. Results show that retrieval-by-decomposition is effective at handling such long and complex queries.

**Reasons To Accept:**

 - A new problem is defined: retrieve book by user's long and complex description. A new dataset, WhatsThatBook, is built for this problem.
 - A new approach is proposed: retrieval-by-decomposition. The proposed approach is a novel application of large language models (LLM). The new approach outperforms a few strong baselines in the empirical evaluation.
 - The paper is easy to follow. Readers should be able to reproduce the results without a lot of difficulties.


**Reasons To Reject:**

The number of documents (i.e., the book to find) is rather small in the constructed dataset (only about 14k documents in total). The experiments are conducted on these documents only. It is unknown if the results will generalize if there are more candidate books in the knowledge base, which is a more realistic setting.

**Reproducibility:**

4: Could mostly reproduce the results, but there may be some variation because of sample variance or minor variations in their interpretation of the protocol or method.

**Reviewer Confidence:**

4: Quite sure. I tried to check the important points carefully. It's unlikely, though conceivable, that I missed something that should affect my ratings.

---

> ### Author Rebuttal · Authors · 2023-08-29
>
> We appreciate that the reviewer found the formulation of the problem to be interesting, and our approach of retrieval-by-decomposition to be novel.
>
> We completely agree with the reviewer that a larger dataset of more books would be more realistic, and to this end, we have collected an additional 172,422 negative books (books that do not correspond to a gold query in our dataset) to make the setting more realistic. This has given us a total of 14,441 queries and 186,863 documents. With all documents added to the corpus, our contriever baseline achieves:
>
> Recall at 5 = 12.2
>
> Recall at 10 = 16.1
>
> Recall at 20 = 20.1
>
> Recall at 100 = 33.4
>
> while our predicted clues and full decomposition gets:
>
> Recall at 5 = 18.6
>
> Recall at 10 = 23.4
>
> Recall at 20 = 28.6
>
> Recall at 100 = 43.5.
>
> In comparing with our results in Table 4, observe: (1) All model performances are lower under the setting with more negative books, as the task is now more difficult showing that there is more headroom on the task, and (2) our approach is still outperforming the contriever baseline. If accepted, we will include these results in our extra page for the camera-ready and additionally release our collected set of negative books to support this type of experimentation.
>
> On the topic of dataset size, we believe our dataset is still valuable. Indeed, there exist larger datasets like MSMarco or NaturalQuestions, but we also believe there is a need to complement these datasets with others that allows researchers to study retrieval systems under different types of queries or documents. In our case, that target phenomenon of interest are the long, complex queries that arise naturally from tip-of-the-tongue settings. Taking the popular BEIR benchmark datasets [1], for example, the only dataset that has comparable query length and complexity is ArguAna [2], which contains 1,406 queries and 8,674 documents, far smaller than our dataset. We hope this helps increase the reviewer’s confidence in our contributions to this research community.
>
> [1] https://arxiv.org/abs/2104.08663
>
> [2] https://aclanthology.org/P18-1023.pdf

---

### Meta-Review · Area_Chair_CR94 · 2023-09-18

**Recommendation:** 3

**Metareview:**

This paper is focused on developing resources and methods for complex retrieval tasks. Towards that end, they have created a new dataset which involves long text queries based on real users trying to remember the identity of particular books (e.g., describing the cover, along with vague details about the author or story). The paper also presents a new method for this task, based on decomposing the query, and compares against classical IR baselines.

Reviewers found this paper to be interesting, novel, and easy to follow. The new dataset was seen as potentially useful to the community.

The two main concerns were that 1) the methodology was somewhat ad hoc, and left room for improvement, both in terms of rigor and sophistication, and 2) limitations with the evaluation, including the use of only a single dataset, lack of ablations, and lack of more recent baselines for comparison. In general, it seems like there is a concern that although the authors have demonstrated an effective method, it is unclear exactly why it works, or how well it will generalize.

In their rebuttal, the authors have included additional results, for both additional baselines and additional datasets. Although reviewers have acknowledged the rebuttal, they have unfortunately not specifically engaged with the new results provided by the authors. Given the difficulty of assessing out of context results provided in a rebuttal, this is not terribly surprising, and potentially reveals a problem with encouraging authors to include new experimental results in their rebuttal.

Overall, this seems like a case where this paper could benefit from an additional round of revisions before publication. However, all reviewers rated this paper as good or better on soundness, so they nevertheless felt it was worthy of being accepted, even if the paper's limitations tempered their excitement.

---

### Decision · Program_Chairs · 2023-10-07

**Decision:**

Accept-Findings

**Comment:**

This paper is focused on developing resources and methods for complex retrieval tasks. Towards that end, they have created a new dataset which involves long text queries based on real users trying to remember the identity of particular books (e.g., describing the cover, along with vague details about the author or story). The paper also presents a new method for this task, based on decomposing the query, and compares against classical IR baselines.

Reviewers found this paper to be interesting, novel, and easy to follow. The new dataset was seen as potentially useful to the community.

The two main concerns were that 1) the methodology was somewhat ad hoc, and left room for improvement, both in terms of rigor and sophistication, and 2) limitations with the evaluation, including the use of only a single dataset, lack of ablations, and lack of more recent baselines for comparison. In general, it seems like there is a concern that although the authors have demonstrated an effective method, it is unclear exactly why it works, or how well it will generalize.

In their rebuttal, the authors have included additional results, for both additional baselines and additional datasets. Although reviewers have acknowledged the rebuttal, they have unfortunately not specifically engaged with the new results provided by the authors. Given the difficulty of assessing out of context results provided in a rebuttal, this is not terribly surprising, and potentially reveals a problem with encouraging authors to include new experimental results in their rebuttal.

Overall, this seems like a case where this paper could benefit from an additional round of revisions before publication. However, all reviewers rated this paper as good or better on soundness, so they nevertheless felt it was worthy of being accepted, even if the paper's limitations tempered their excitement.